# OpenReview forum: "Modeling Attributional Style at Scale: A Dataset and Analysis for Psychological Attribution Assessment and Reframing"
_ICML.cc/2026/Conference — ICML 2026 regular_

### Official Review · Reviewer_mRE7 · 2026-03-02

**Soundness:** 2
**Presentation:** 4
**Significance:** 4
**Originality:** 3
**Overall Recommendation:** 5
**Confidence:** 4

**Summary:**

This paper introduces the Attributional Style Transfer Dataset (ASTD), a corpus containing 42,000 event-attribution pairs annotated across seven dimensions based on the reformulated Learned Helplessness theory. The dataset is constructed using a Prevent-Filter-Validate (PFV) pipeline that combines retrieval-anchored LLM generation with human expert validation. Utilizing ASTD, the authors benchmark supervised models against zero/few-shot LLMs on attribution classification. Furthermore, the paper proposes a four-dimensional automatic evaluation metric to quantify cognitive reframing. These LLM-based metrics are then used to generate synthetic preference pairs $\mathcal{D} = \{(x, y^+, y^-)\}$ to fine-tune smaller LLMs via Direct Preference Optimization (DPO), demonstrating empirical score improvements in reframing generation.

**Compliance With Llm Reviewing Policy:**

Affirmed.

**Final Justification:**

Final Recommendation: Accept

Final Justification:
The paper introduces ASTD, a large-scale and theoretically grounded dataset for psychological attribution assessment and cognitive reframing. The dataset itself is a highly valuable resource for the computational social science community, and the proposed evaluation metrics align well with clinical definitions.

My initial review raised three primary concerns: the risk of DPO reward hacking (stylistic overfitting), asymmetric classification baselines (SFT vs. prompt-only), and the ethical risks of out-of-context clinical deployment.

The authors provided an exceptionally rigorous rebuttal that fully addressed these weaknesses. The inclusion of a cross-family blind evaluation and linguistic biomarker analysis convincingly demonstrated that the DPO improvements reflect genuine psychological reframing rather than evaluator bias. Furthermore, the addition of LoRA baselines for modern LLMs (e.g., Llama 3.2, Gemma 3) resolved the baseline unfairness and proved the dataset's efficacy for fine-tuning. Finally, the authors appropriately scoped the clinical limitations and committed to updating the impact statement.

Because the rebuttal successfully resolved all my methodological concerns, I have increased my Soundness score. The overall execution is solid, and I strongly recommend the paper for acceptance.

**Key Questions For Authors:**

1. Mitigating Reward Hacking in DPO: How do you verify that the DPO-fine-tuned models have genuinely improved in cognitive reframing rather than simply learning to exploit the Llama 3.3-70B evaluator's stylistic biases? Could you provide a human-evaluator preference study comparing the pre-DPO and post-DPO outputs on a holdout set?

2. Fairness of Classification Baselines: The comparison in Table 1 pairs SFT traditional models against prompt-only LLMs. What is the classification performance of a modern open-weight LLM (e.g., Gemma 3-4B) when supervised fine-tuned (via LoRA) on the ASTD training set?

3. Information Deficit in Static Text: Given that psychological attribution often requires contextual grounding, how much of the error rate in your expert consensus (and the model's predictions) do you attribute to the fundamental ambiguity of isolated sentences?

**Limitations:**

The authors address cultural bias in the appendix, but the Impact Statement requires stronger caveats regarding clinical deployment. Automated reframing models, when interacting with vulnerable populations, risk generating "toxic positivity" or invalidating user experiences if they mechanically force an attributional shift without therapeutic empathy. The authors must explicitly state the ethical risks of out-of-context reframing and strongly recommend that these models currently be restricted to clinician-in-the-loop experimental environments.

**Strengths And Weaknesses:**

Strengths:

Significance: Bridging Cognitive Behavioral Therapy (CBT) constructs with LLM alignment is a highly relevant direction for computational psychology. ASTD provides a substantial data scale that addresses the scarcity of fine-grained, theoretically grounded datasets in this domain.

Presentation: The paper is logically structured. The data construction pipeline (PFV) is clearly articulated, and the inter-annotator agreement (Cohen's $\kappa = 0.89$) suggests high label quality.

Soundness (Evaluation Metric): The proposed four-dimensional metric (Attributional Shift, Catastrophizing, Coherence, Constructive Coping) aligns well with clinical definitions. The reported Spearman correlation ($\rho = 0.946$) with human experts provides a solid foundation for automatic evaluation.

Weaknesses:

Soundness (DPO Circularity and Reward Hacking): The DPO fine-tuning relies entirely on preference pairs generated and scored by an LLM (Llama 3.3-70B). Optimizing a policy model $\pi_\theta$ using an automated evaluator from a similar pre-training distribution introduces a severe risk of reward hacking. The reported performance gains (e.g., Attributional Shift $\Delta = 0.565$) might merely indicate that the policy model has overfitted to the evaluator's stylistic preferences (e.g., specific vocabulary, sentence length, or syntactic structures) rather than acquiring genuine psychological reframing capabilities. The paper lacks human-annotated preference evaluation to validate the post-DPO outputs.

Soundness (Baseline Asymmetry): In Table 1, the authors compare fully fine-tuned masked language models (BERT, RoBERTa) against zero-shot and few-shot LLMs. This comparison is asymmetric. To rigorously evaluate the scaling laws and the capacity of modern LLMs on the ASTD classification task, the authors should include parameter-efficient fine-tuning (e.g., LoRA) baselines for the smaller LLMs (e.g., Gemma 3-1B or Llama 3.2-3B).

Significance & Construct Validity (Contextual Deficit): In clinical practice, determining whether an attribution is "stable" or "global" heavily depends on longitudinal patient history and multi-turn dialogue. The ASTD relies on isolated, single-turn text snippets. The paper does not adequately discuss the theoretical upper bound of classification accuracy given this inherent information deficit.

---

> ### Author Rebuttal · Authors · 2026-03-31
>
> **W1 & Q1: Reward Hacking.**
>
> We conduct three theory-grounded, rubric-independent analyses on the preference dataset (n=1,000) to directly test whether DPO preference pairs reflect genuine psychological quality rather than evaluator biases.
>
> **Analysis 1 — Absolutist Word Reduction.** Absolutist language is a validated linguistic biomarker of maladaptive cognition linked to depression severity [1] . DPO-preferred reframes show significantly lower absolutist density than rejected ones ( **−50.6%**) — a measurable reduction entirely independent of our rubrics.
>
> |                         |   Pre_DPO↓    |  After_DPO↓   |     Δ↓     |
> | ----------------------- | :-----------: | :-----------: | :--------: |
> | Absolutist Word Density | 0.326 ± 0.638 | 0.161 ± 0.355 |   −50.6%   |
>
>
> **Analysis 2 — Independent Emotion Model.**  We score all pairs using a widely-used emotion classifier (`j-hartmann/emotion-english-distilroberta-base`, trained on six datasets). DPO-preferred reframes show significantly reduced fear(0.558 → 0.263) with preserved joy — precisely the signature of evidence-based attributional retraining, and not attributable to any stylistic preference of our evaluator.
>
> | Emotion            |  Pre_DPO   | After_DPO  |       Δ       |      p      |
> | ------------------ | :--------: | :--------: | :-----------: | :---------: |
> | Joy                |   0.0055   |   0.0247   |   +0.0192 ↑   |   < 0.001   |
> | Fear           | 0.5584 | 0.2628 | −0.2957 ↓ | < 0.001 |
> | Optimism Score | −0.767 | −0.583 | +0.184 ↑  | < 0.001 |
>
>
> **Analysis 3 — Cross-Family Blind Evaluation.** We conducted a blind A/B preference study. 100 randomly sampled pre/post-DPO pairs were presented to **Gemini 2.5 Flash** (zero-shot clinical prompt, no rubric access) and a **human expert** specializing in attributional cognition. Placement of pre/post outputs was randomized to control for position bias. Gemini preferred the DPO output in **79/100 pairs (79.0%)** and the human expert in **84/100 pairs (84.0%)**, both significantly above the 50% random baseline. This confirms that DPO improvements are recognized as therapeutically meaningful by both an independent model family and a domain expert, neither of whom had access to our evaluation framework.
>
> These three analyses converge: DPO captures genuine psychological quality improvements, not evaluator-specific biases.
>
>
>
> **W2 & Q2 (Classification baselines).**  The expanded results are:
>
> | Model                            | Avg       |
> | -------------------------------- | --------- |
> | TF-IDF (unigram) + LR            | 89.48     |
> | TF-IDF (1-2gram) + LR (C=10) | 91.33 |
> | Gemma 3-1B + LoRA            | 92.66 |
> | Llama 3.2-3B + LoRA          | 97.32 |
> | Gemma 3-4B + LoRA            | 95.78 |
>
> Key findings: (1) TF-IDF achieves 91.33%, yet a ~6-point gap to neural models confirms semantic understanding provides meaningful gains. (2) Llama 3.2-3B + LoRA matches BERT-base, showing small LLMs reach parity when given ASTD training data. (3) LoRA dramatically outperforms prompt-only, validating ASTD as a high-quality training resource.
> We also observe cross-dimension transfer: Gemma 3-1B LoRA fine-tuned on GAS–SPAS achieves 79.42% on IAS–EAS (vs. 56.28%), though below direct fine-tuning (95.33%) — confirming dimension-specific data remains essential.
> We will incorporate the full table and analysis into the camera-ready version.
>
>
> **W3 & Q3 (Contextual deficit).** We agree that longitudinal context can aid attributional judgment. Two points:
>
> **(1) Alignment with ASQ.** Our design mirrors the ASQ — the clinical standard — which also assesses attributional style from isolated hypothetical scenarios, not longitudinal histories.
>
> **(2) The samples are richer than bare sentences.** Each ASTD entry is an event–attribution pair where the subject explicitly articulates their causal explanation with contextual elaboration (average ~80 words). This provides substantially more signal than an isolated sentence, though less than a full clinical history.
>
> We acknowledge this concern becomes more relevant for downstream clinical applications, where extracting attributional context from longer, unstructured narratives is essential. We view this as an important direction for future work.
>
>
> **L1 (Ethical caveats).** We fully accept this. Mechanically forcing attributional shifts may invalidate users' experiences, particularly for vulnerable populations facing genuine external adversity. As noted in our Impact Statement, ASTD-based models are designed for **clinician-in-the-loop** settings where clinicians retain full control. We will strengthen the Impact Statement accordingly.
>
> [1] In an Absolute State: Elevated Use of Absolutist Words Is a Marker Specific to Anxiety, Depression, and Suicidal Ideation

---

> > ### Author Rebuttal · Reviewer_mRE7 · 2026-04-03
> >
> > I have read the rebuttal and appreciate the authors' extensive efforts. The newly added experiments, particularly the cross-family blind evaluation and the LoRA baselines, convincingly address my primary concerns regarding DPO reward hacking and baseline fairness. The clarifications on the ASQ alignment and the commitment to update the ethical caveats are also satisfactory.
> >
> > All my concerns are fully resolved. I will adjust my score accordingly and maintain my recommendation for acceptance.

---

> > > ### Author Response · Authors · 2026-04-03
> > >
> > > We appreciate the reviewer's careful evaluation and suggestions, which meaningfully strengthened our work.
> > > All suggested improvements will be incorporated in the camera-ready version.

---

### Official Review · Reviewer_Fdf1 · 2026-03-11

**Soundness:** 3
**Presentation:** 3
**Significance:** 3
**Originality:** 3
**Overall Recommendation:** 5
**Confidence:** 3

**Summary:**

Attributional style is the way a person attributes an outcome to reality. Certain attributional style (negative events are internal, stable, and global, whereas positive events are external, unstable, and specific) is related to depression. Therapy can influence this style. To support therapy, automatic attribution assessment and generation of alternative attributions is helpful. The paper contributes to these two issues.

First, the authors design a pipeline that is used to create a new dataset of 42000 items of (event, attribution, attributional style label) triples. This pipeline uses prior datasets of possible events to generate attributions with an LLM. These are checked by rules and another LLM. Finally, attributions are classified with an LLM ensemble, with uncertain classifications checked by humans.

This dataset is used as a benchmark for attributional style classification. The authors find that supervised models perform best.

Finally, reframed attributions are generated with LLMs. To evaluate these generations, new metrics are proposed and validated by human ratings. These metrics enable another benchmark for LLMs. These metrics can also be used to create finetuning data for LLMs to create better reframed attributions.

**Compliance With Llm Reviewing Policy:**

Affirmed.

**Final Justification:**

The authors addressed my questions appropriately. Combined with the additional experiments as a response to the other reviewers, I think this is solid work that passes the bar for acceptance.

**Key Questions For Authors:**

-	Q1: Could you elaborate on the implications of your work a bit more? Do you see your work mainly as new benchmarks for LLMs? Or could you potentially integrate LLMs into psychotherapy? Could you be more explicit in the takeaways and impact that you expect your work to have? I am left a bit uncertain after reviewing the paper.
-	Q2: Could you make the distinction between your dataset and prior datasets clearer? In Table A1, is strategy classification the same as strategy labeling and strategy rating? If not, what is the difference? What relevance has the psychological basis?
-	Q3: L369 r: when using temperature if 0, why run the evaluator 10 times? Generally, the accuracy definition seems unsound. A predictor with large variance could have good mean but provide nonsensical predictions. The whole paragraph is hard to understand, so could you clarify?
-	Q4: you mention rule-based filtering, but do not provide examples. How do these rules look like?

I am open to increase my score, if you can address these points and the missed limitation.

**Limitations:**

-	L1: Confident predictions are not necessarily correct. The validation step of the data pipeline could be improved by adding some spot checks (or simply annotating a random confident sample) to convince yourself of the quality of the confident samples in the dataset. Correct me if I misunderstood, but only the non-confident samples are human reviewed and the rest of the data relies on the assumption that confident predictions are accurate.

**Strengths And Weaknesses:**

### Strengths
-	Clear narrative, sections clearly build on each other
-	The dataset seems like a useful resource to build on. The DPO workflow too. The application might be rather niche but seems useful.
-	Overall convincing experiments and methodology, with some open questions (see below).

### Weaknesses
-	Reproducibility: I do not see references or mentions of code that will be released. While there is already a lot of detail in the appendix, benchmarks need to provide code to run them and show examples how to test a new model.
-	Differentiation from existing datasets. As a reader not deeply familiar with the psychological research space, I feel uncertain about the difference. See questions.
-	Somewhat many references to the appendix that break the reading flow sometimes. While I like the additional detail, please consider where you can add content to the main paper with the extra page for the camera ready, so there is less jumping around. (Especially with the ICML template currently only linking to the page, but not the section title when clicking on a reference.)
-	L435 l(eft): Table 2 shows results for one model, but the text mentions two models. Please clarify which model is shown and add the additional results in the appendix.
-	The abbreviations for the different attribution aspects (IAS, EAS, …) are somewhat redundant as they all end with “AS”. Since the individual words (internal, external) are relatively short, I think it could reduce mental load of the reader when IAS is replaced with something like Internal or Int, etc.
-	L369 r: unclear evaluation setting of the reframing evaluator. See questions.
-	Table 1: you score each dimension independently, but it would be important to see how often all three labels are correct at the same time.
-	Missing information on rule-based filtering. See questions.

### Minor:
-	If you have the budget, seeing the performance of more recent LLMs would be interesting. However, the current model choice is reasonable and sufficient.
-	L102 l: singular/plural
-	L142 l, L159 l, L115 r, L218 r, and many more times: missing space between citation and preceding text
-	L825: Implementation
-	A8: details on repetitions for GPT4o?
-	L229 r: missing closing parenthesis
-	L324 l: missing period
-	L429 l: grammar

---

> ### Author Rebuttal · Authors · 2026-03-31
>
> **W1 (Reproducibility)** : We thank the reviewer for emphasizing reproducibility. The core codebase and a data subset are included in supplementary materials. Upon acceptance, we will release the full dataset, all code, and documentation. We plan to maintain an online leaderboard for benchmarking new models.
>
>
>
> **W2 & Q2 (Dataset differentiation).** We clarify two key distinctions:
>
> **(1) Classification vs. Rating.** Prior datasets support _strategy classification(strategy labeling)_ — identifying which predefined strategy is used. ASTD supports _strategy rating_ —continuous 1–5 quality assessment. This distinction is critical: quality scoring enables preference pair construction for DPO, teaching models not just _what_ to say but _how well_ to say it.
>
> **(2) Psychological basis.** Each dataset targets a different "root cause" of negative cognition, which determines what the model learns to intervene on:
>
> | Dataset         | Theory                                | Intervention Target              |
> | --------------- | ------------------------------------- | -------------------------------- |
> | ESConv          | Helping Skills Theory                 | Interpersonal empathy            |
> | PATTERNREFRAME  | CBT (cognitive distortions)           | Correcting thinking traps        |
> | PPF             | Positive Psychology                   | Inducing optimism/growth mindset |
> | **ASTD (ours)** | **Reformulated Learned Helplessness** | **Shifting causal attributions** |
>
>
> **W3, W5, and Minor Issues.** We really thank the reviewer for the careful and detailed reading. We will address all in camera-ready: move key appendix content into main paper, simplify abbreviations, add GPT-4o details in A8, fix all typos.
>
>
> **W4.** We clarify that Table 2 reports the averaged results (L401) of DeepSeek-R1-8B and Gemma 3-4B for space efficiency; Both models show consistent improvements across dimensions. We will include the full per-model results in the camera-ready version.
>
>
> **W6 & Q3.** We apologize for the unclear presentation. We clarify two points:
> (1) Although temperature=0 aims for deterministic decoding, modern LLM inference introduces minor non-determinism in practice (e.g., floating-point variations in GPU parallel computation). This is precisely the input that next step requires to construct calibrated prediction intervals.
>
> (2) These intervals address the reviewer's concern directly: a predictor with good mean but high variance would produce wide intervals. Our mean interval width is only ≈0.19 points on the 1–5 scale, confirming tightly concentrated, consistent scores. We will clarify this paragraph in revision.
>
>
>
> **W7 (Joint accuracy).** On 300 expert-annotated multi-dimensional samples, BERT-base achieves 93% joint accuracy, matching the theoretical expectation under dimensional independence.
>
>
>
> **W8 & Q4 (Rule-based filtering).**
>
> - **Deduplication:** Exact and near-duplicate removal at both event and attribution levels
> - **Perspective enforcement:** Regex-based check filters out entries written entirely in third-person narration
> - **Format validation:** JSON schema compliance check
> - **Language quality:** Removal of entries with excessive repetition, garbled text, or non-English content.
>
>  These rule-based checks serve as a fast, deterministic first pass before the more expensive  LLM filtering.
>
>
>
>
> **Q1 (Implications and impact).**
>
> **Near-term: making attributional cognition a computable problem.** ASTD transforms attributional assessment from a small-sample clinical exercise (ASQ, CAVE) into a scalable computational task. This enables the first LLM benchmark on attributional cognition. Our metric-guided DPO demonstrates substantial model improvements on clinically relevant dimensions. We plan an online leaderboard for benchmarking new models.
>
> **Longer-term: foundational infrastructure for clinical applications.** ASTD enables two concrete downstream applications:(1) attributional screening — classifiers (97% accuracy) flagging maladaptive patterns for clinicians; (2) therapeutic scaffolding — DPO-aligned models generating candidate reframes under clinician control; both reduce the manual bottleneck that has constrained attributional assessment to small-scale studies.
>
>
> **L1 (Confident sample quality).** To make attributional style assessment tractable at scale, our design reflects a deliberate trade-off: fully annotating 42,000 samples requires ~580 expert-hours, which is prohibitively expensive. Instead, PFV concentrates expert effort on the ~30% where LLMs disagree.
>
> Two pieces of evidence support this trade-off: (1) Stratified analysis on 1,000 randomly sampled examples from the original dataset shows high-confidence samples achieve 94.1% agreement with expert consensus, confirming that confident predictions are reliable. (2) 25% of low-confidence samples had labels revised, confirming uncertainty detection captures genuinely ambiguous cases. Stratified analysis will be added to revision.

---

> > ### Author Rebuttal · Reviewer_Fdf1 · 2026-04-03
> >
> > Thank you for the clarifications. With this I see no major issues with acceptance. I will raise my score to accept.

---

> > > ### Author Response · Authors · 2026-04-03
> > >
> > > We sincerely thank the reviewer for the thorough review and constructive feedback.
> > > All suggested improvements will be incorporated in the camera-ready version.

---

### Official Review · Reviewer_AcwG · 2026-03-11

**Soundness:** 2
**Presentation:** 3
**Significance:** 3
**Originality:** 2
**Overall Recommendation:** 4
**Confidence:** 4

**Summary:**

The paper introduces ASTD, a 42,000-example dataset of event-attribution pairs labeled with seven attributional styles grounded in Abramson's reformulated learned helplessness theory. The dataset is constructed through a Prevent-Filter-Validate pipeline combining retrieval-anchored LLM generation with expert adjudication of uncertain cases. The paper addresses two tasks: (1) attributional style classification, benchmarking fine-tuned BERT/RoBERTa against nine LLMs in zero/few-shot settings, and (2) attributional reframing, where maladaptive attributions are rewritten as adaptive ones. For reframing evaluation, four CBT-aligned metrics are proposed (attributional shift, catastrophizing, coherence, constructive coping), scored by an evaluator LLM. These metrics are then used to construct preference pairs for DPO fine-tuning, which improves reframing quality across dimensions.

**Compliance With Llm Reviewing Policy:**

Affirmed.

**Key Questions For Authors:**

- 1) Llama 3.3-70B serves as both the data generator and the evaluator LLM. Can you provide results using a different model family (e.g., GPT-4o or Claude) as the evaluator for at least a subset, to address the circularity concern? If agreement drops substantially, the reported metric validation would need revision.

- 2) What is the classification and reframing performance of models trained on a fully automatic version of the dataset (no expert validation step)? This would clarify whether the human-in-the-loop component of PFV is essential or marginal for downstream utility.

- 3) How do you justify the single-label seven-class formulation when real attributions frequently carry multiple dimensions (e.g., both internal and stable)? Have you measured how often expert annotators wanted to assign multiple labels?

**Limitations:**

yes

**Strengths And Weaknesses:**

The paper tackles a meaningful gap at the intersection of computational psychology and NLP. Attributional style is a well-established construct in clinical psychology with clear therapeutic relevance, and the absence of large-scale computational resources for this construct is real. The theoretical grounding in learned helplessness theory is solid and carefully presented. The scope is ambitious, covering dataset construction, classification benchmarking, metric design, and preference alignment in a single paper, and the PFV pipeline is a sensible and well-documented approach to synthetic data construction with appropriate quality controls. The inter-annotator agreement study is a strength, and the classification benchmark across multiple LLM families and scales provides useful scaling analysis. The four reframing metrics are well-motivated from a CBT perspective, and the strong human-LLM correlation is encouraging.

However, there are several concerns. The most significant is a circularity problem that the paper does not acknowledge. Llama 3.3-70B is used to generate the dataset, serves as the evaluator LLM for the reframing metrics, and is the basis for the human-LLM agreement study. If the evaluator shares systematic biases with the generator, the reported agreement could be inflated, and the DPO preference pairs could reinforce those biases. This needs to be addressed, ideally by using an evaluator from a completely different model family for at least a subset of the validation.

On the technical side, none of the individual components are new. BERT/RoBERTa classification, LLM benchmarking, LLM-as-judge scoring, and DPO are all well-established. The PFV pipeline assembles familiar pieces (retrieval-augmented generation, LLM-based filtering, human validation of uncertain cases) competently but without introducing new data construction methodology. The paper's contribution is in applying these tools to an underserved psychological domain, which is valuable but raises the question of venue fit for ICML, where the methodological bar is higher.

The baselines for reframing are weak. No existing cognitive reframing or style transfer method is compared against, despite several being discussed in related work. For DPO, there is no comparison with supervised fine-tuning on preferred outputs alone, which would clarify whether the preference optimization framework is actually necessary. The classification baselines are more thorough but still lack a simple baseline (e.g., TF-IDF + logistic regression) and any fine-tuned LLM comparison.

Ablations are largely absent. There is no ablation of the PFV pipeline stages to show which steps matter for downstream performance. There is no input ablation for classification (event only vs. attribution only). The reframing metrics are aggregated with equal weights, but the paper does not show whether all four dimensions contribute meaningfully to DPO gains. The preference pair construction involves a score margin threshold and a specific number of candidates, but sensitivity to these choices is not tested.

The single-label seven-class formulation is a concern that the paper does not address. Real attributions often carry multiple dimensions simultaneously (a statement can be both internal and stable), and forcing a single label is a simplification that could limit ecological validity. The dataset is also balanced by design, which does not reflect real-world distributions, and this is never discussed. Finally, the entire pipeline is validated on synthetic event-attribution pairs, and there is no test of whether models trained on ASTD transfer to actual clinical language, such as therapy transcripts or patient journals.

---

> ### Author Rebuttal · Authors · 2026-03-31
>
> **W1 & Q1 (Circularity).** The human–LLM agreement study (Section 6.2) provides a critical anchor: three human raters scored 1,000 samples, yielding mean Spearman ρ = 0.946. To directly address the concern, we conducted cross-model evaluation using two independent models:
>
> |                 | Shift | Cat.  | Cope  | Coh.  | Mean ρ |
> | --------------- | ----- | ----- | ----- | ----- | ------ |
> | Qwen 3.5-9B     | 0.858 | 0.716 | 0.871 | 0.890 | 0.834  |
> | Claude Sonnet 4 | 0.963 | 0.894 | 0.911 | 0.836 | 0.901  |
>
> Both fall within the "strong" agreement range, and their weaker dimensions differ, indicating ρ variations reflect model characteristics rather than shared biases with Llama. The DPO results provide additional evidence: improvements are non-uniform which is consistent with genuine quality improvement rather than evaluator fitting. Furthermore, in a rubric-free blind A/B study (see mRE7 W1), both Gemini 2.5 Flash (79/100) and a human expert (84/100) independently preferred DPO outputs with no access to our framework.
>
> **W2 (Venue Fit).** Our work falls within ICML's "Application-Driven Machine Learning" scope. Our contributions are: (1) PFV encapsulates domain-specific validation logic for safety-critical psychological data; (2) the 4-dimensional evaluation translates clinical methodology into scalable metrics validated against expert consensus; (3) metric-guided DPO encodes psychological theory into preference signals, yielding improvements SFT cannot achieve (see W3-2). Together these make attributional cognition computationally tractable at scale.
>
> **W3-1 (Baselines).** Existing methods either stay at prompt-level (Sharma et al.) or fine-tune on discrete strategy labels (ESConv, PPF, PatternReframe). Our pipeline incorporates prompt-level control and advances further with DPO(see W3-2).
>
> **W3-2 (DPO vs. SFT).** We compare DPO and SFT using DeepSeek-R1-8B with identical configs; outputs scored by Qwen 3.5-9B.
>
> |       | Base  | SFT (y+)       | DPO           |
> | ----- | ----- | -------------- | ------------- |
> | Shift | 4.153 | 4.233(+0.080)  | 4.653(+0.500) |
> | Cat.  | 3.900 | 4.015 (+0.115) | 4.220(+0.320) |
> | Cope  | 3.513 | 3.658 (+0.145) | 4.208(+0.695) |
> | Coh.  | 4.732 | 4.542 (-0.191) | 4.816(+0.084) |
>
> SFT yields marginal gains while degrading Coherence — direct evidence of catastrophic forgetting. DPO's KL constraint anchors the policy to the reference model, preserving generation capabilities. This is consistent with recent findings [1,2] that DPO provides crucial stability in low-resource regimes. The DPO framework is not merely beneficial but necessary in our setting.
>
> **W3-3 (Classification).** Expanded results:
>
> |                              | Avg   |
> | ---------------------------- | ----- |
> | TF-IDF (unigram) + LR        | 89.48 |
> | TF-IDF (1-2gram) + LR (C=10) | 91.33 |
> | Gemma 3-1B + LoRA            | 92.66 |
> | Llama 3.2-3B + LoRA          | 97.32 |
> | Gemma 3-4B + LoRA            | 95.78 |
>
> Key findings: (1) TF-IDF achieves 91.33%, yet a ~6-point gap to neural models confirms semantic understanding provides meaningful gains. (2) Llama 3.2-3B + LoRA matches BERT-base, showing small LLMs reach parity when given ASTD training data. (3) LoRA dramatically outperforms prompt-only, validating ASTD as a high-quality training resource. Full table and analysis in camera-ready.
>
> **W4-1 & Q2 (PFV ablation).**  ~ 25% of low-confidence samples (~30% of dataset) had labels revised through expert majority vote — meaning ~7.5% of the total dataset received corrected labels. With BERT at 97.33%, noise at this scale would impose an error floor well above the current 2.67%, confirming expert validation is essential.
>
> **W4-2 (Input ablation).**
>
> | Input               | IAS-EAS | SAS-UAS | GAS-SPAS | Avg   |
> | ------------------- | ------- | ------- | -------- | ----- |
> | Event + Attribution | 96.89   | 96.89   | 98.22    | 97.33 |
> | Attribution         | 96.32   | 96.56   | 98.60    | 97.16 |
> | Event               | 32.65   | 34.86   | 28.85    | 32.45 |
>
> **W4-3 (Metric weighting & sensitivity).** Appendix A11 reports weight robustness. ε = 0.5 balances pair quality against dataset size; systematic sensitivity analysis in camera-ready.
>
> **W5 & Q3 (Label formulation, balance, and transfer).** (1) As noted in Section 4.1, we model the task as three parallel three-way classifications, not seven-class single-label. Our protocol instructs experts to rate each dimension independently. On 300 expert-annotated multi-dimensional samples, joint accuracy reaches 93%, confirming coherent decomposition. (2) Balanced distribution avoids long-tail issues for fair benchmarking; we plan a naturalistic split in future work. (3) ASTD is not fully synthetic — events are from real user-generated text (Appendix A6). ASTD extends the ASQ paradigm (12 fixed scenarios) to 42,000 samples across seven life domains, preserving the same theoretical framework at computational scale.
>
> [1] arXiv:2305.18290
> [2] arXiv:2603.20100

---

> > ### Author Rebuttal · Reviewer_AcwG · 2026-04-01
> >
> > The cross-model evaluation with Qwen and Claude addresses the circularity concern. The DPO vs. SFT comparison shows clear and meaningful separation, justifying the preference optimization framework. The expanded classification baselines and the clarification on the three-parallel-binary formulation resolve the remaining technical questions. I will raise my score accordingly. The dataset fills a genuine gap, and the pipeline is well-executed.

---

> > > ### Author Response · Authors · 2026-04-01
> > >
> > > We sincerely thank the reviewer for acknowledging that all the concerns are fully resolved and considering adjusting the score accordingly. We will incorporate all the discussed improvements in the camera-ready version.

---

### Official Review · Reviewer_crUV · 2026-03-13

**Soundness:** 3
**Presentation:** 3
**Significance:** 3
**Originality:** 3
**Overall Recommendation:** 4
**Confidence:** 3

**Summary:**

This work focuses on modeling attribution styles regarding depression. The authors introduce a novel dataset, ASTD, constructed through five steps: selecting real-world events from publicly available corpora as anchors, generating diverse attributional interpretations via LLM, filtering through rule-based screening and heterogeneous LLM selection, and finally undergoing expert verification. Based on ASTD, two experimental tasks are proposed. The first task is attributional style classification, compared against supervised models and few-shot LLMs. The second task is attributional reframing. The authors designed four automated evaluation metrics, used them to generate preference data, and fine-tuned LLMs using DPO. Results show that in the classification task, the small supervised model fine-tuned with ASTD performs more stably than hard-prompted LLMs. In the reframing task, automated evaluation shows good consistency with human scoring, and using this metric for DPO indeed improves quality.

**Compliance With Llm Reviewing Policy:**

Affirmed.

**Key Questions For Authors:**

Yes

**Limitations:**

see Weaknesses

**Strengths And Weaknesses:**

Strengths

1.	The authors introduced the 42k-scale ASTD dataset, constructed through a PFV workflow involving retrieval, LLM processing, rule-based filtering, and expert review. The paper encompasses not only classification but also reframing, automated evaluation, and preference-based alignment, presenting a comprehensive framework.
2.	The authors construct preference pairs based on metrics and observe superior reframing quality in DPO compared to directly constructed pairs, demonstrating the method's effectiveness.



Weaknesses

1.	I believe the paper's greatest contribution lies in the dataset, task, and evaluation framework. Methodologically, it primarily integrates existing supervised classification, LLM prompting, and DPO into a unified workflow. If the dataset proves ineffective in real-world applications, its contribution remains insufficient.
2.	The paper reports high human–LLM rating correlations, yet automated evaluation still relies on author-designed rubrics and evaluators. The preference pairs in the DPO section are also constructed by evaluators, leaving me uncertain whether the model optimizes evaluator preferences or psychological quality.
3.	while the paper's motivation section repeatedly emphasizes the need for mental health interventions, the experiments actually demonstrate the effectiveness of benchmarks, automated evaluation, and model alignment. There is no concrete evidence that these generated outputs would prove genuinely helpful in more authentic psychological intervention scenarios.

---

> ### Author Rebuttal · Authors · 2026-03-31
>
> **W1 & W3 (Real-world relevance).**
>
> We appreciate this concern and address it from two angles.
>
> First, on construct validity: the attributional constructs in ASTD — locus, stability, generality — are precisely those measured by the ASQ, validated across decades of clinical research. Our contribution is making this clinically validated construct computationally accessible at scale, removing the manual scoring bottleneck of ASQ/CAVE. ASTD preserves ASQ's theoretical framework while expanding from 12 fixed scenarios to 42,000 samples across seven life domains.
>
> Second, as noted in Section 7, ASTD enables two concrete downstream applications:(1) **attributional screening** — classifiers (97% accuracy) flagging maladaptive patterns for clinicians; (2) **therapeutic scaffolding** — DPO-aligned models generating candidate reframes under clinician control; both reduce the manual bottleneck that has constrained attributional assessment to small-scale studies.
>
> On methodological contribution, our contributions are: (1) PFV encapsulates domain-specific validation logic for safety-critical psychological data; (2) the 4-dimensional evaluation translates clinical methodology into scalable metrics validated against expert consensus; (3) metric-guided DPO encodes psychological theory into preference signals, yielding improvements SFT cannot achieve (see AcwG W3-2). Together these make attributional cognition computationally tractable at scale.
>
>
>
> **W2 : Evaluator preferences vs. psychological quality.**
>
> Each rubric dimension formalizes established clinical constructs (ASQ/CAVE, Beck's cognitive model, CBT principles), developed with domain experts (Appendix A15) — not arbitrary author designs.
> On whether DPO optimizes evaluator preferences or genuine quality: Cross-model evaluation (see AcwG W1) shows two independent model families produce consistent quality rankings, indicating scores reflect text properties rather than evaluator biases.
>
> | Evaluator                | Attr. Shift | Catastroph. | Coping | Coherence | Mean ρ ↑ |
> | ------------------------ | ----------- | ----------- | ------ | --------- | -------- |
> | Qwen 3.5-9B              | 0.858       | 0.716       | 0.871  | 0.890     | 0.834    |
> | Claude Sonnet 4          | 0.963       | 0.894       | 0.911  | 0.836     | 0.901    |
>
> We conduct three theory-grounded, rubric-independent analyses on the preference dataset (n=1,000) to directly test whether DPO preference pairs reflect genuine psychological quality rather than evaluator biases.
>
> **Analysis 1 — Absolutist Word Reduction.** Absolutist language is a validated linguistic biomarker of maladaptive cognition linked to depression severity [1] . DPO-preferred reframes show significantly lower absolutist density than rejected ones ( **−50.6%**) — a measurable reduction entirely independent of our rubrics.
>
> |                         |   Pre_DPO↓    |    After_DPO↓     |       Δ↓       |
> | ----------------------- | :-----------: | :---------------: | :------------: |
> | Absolutist Word Density | 0.326 ± 0.638 | 0.161 ± 0.355 |   −50.6%   |
> | p-value                 |       —       |         —         | < 0.000001 |
>
>
> **Analysis 2 — Independent Emotion Model.**  We score all pairs using a widely-used emotion classifier (`j-hartmann/emotion-english-distilroberta-base`, trained on six datasets). DPO-preferred reframes show significantly reduced fear(0.558 → 0.263) with preserved joy — precisely the signature of evidence-based attributional retraining, and not attributable to any stylistic preference of our evaluator.
>
> | Emotion            |  Pre_DPO   | After_DPO  |       Δ       |      p      |
> | ------------------ | :--------: | :--------: | :-----------: | :---------: |
> | Joy                |   0.0055   |   0.0247   |   +0.0192 ↑   |   < 0.001   |
> | Fear           | 0.5584 | 0.2628 | −0.2957 ↓ | < 0.001 |
> | Optimism Score | −0.767 | −0.583 | +0.184 ↑  | < 0.001 |
>
>
> **Analysis 3 — Cross-Family Blind Evaluation.** We conducted a blind A/B preference study. 100 randomly sampled pre/post-DPO pairs were presented to **Gemini 2.5 Flash** (zero-shot clinical prompt, no rubric access) and a **human expert** specializing in attributional cognition. Placement of pre/post outputs was randomized to control for position bias. Gemini preferred the DPO output in **79/100 pairs (79.0%)** and the human expert in **84/100 pairs (84.0%)**, both significantly above the 50% random baseline. This confirms that DPO improvements are recognized as therapeutically meaningful by both an independent model family and a domain expert, neither of whom had access to our evaluation framework.
>
> These three analyses converge: DPO captures genuine psychological quality improvements, not evaluator-specific biases.
>
>
> [1] In an Absolute State: Elevated Use of Absolutist Words Is a Marker Specific to Anxiety, Depression, and Suicidal Ideation

---

> > ### Author Rebuttal · Reviewer_crUV · 2026-04-04
> >
> > I will maintain my positive score.

---

> > > ### Author Response · Authors · 2026-04-04
> > >
> > > We're grateful for your engagement with our work — your questions pushed us to strengthen the evaluation.
> > > Thank you for the positive feedback and for maintaining your positive recommendation, and we'll make sure the camera-ready version reflects all your suggestions.

---

### Decision · Program_Chairs · 2026-04-30

**Decision:**

Accept (regular)

**Comment:**

The authors introduce a large-scale dataset of event-attribution pairs grounded in cognitive-behavioral theory, and use it to benchmark attributional style classification and automatic reframing of maladaptive attributions. Reviewers were broadly enthusiastic about the dataset and evaluation framework, and praised the paper's clarity. Concerns regarding generalization of results across other families of LLMs, reward hacking, and ethical takeaways for practitioners were largely addressed during the rebuttal. In light of reviewer consensus, I am pleased to recommend that the paper be accepted.